# Immune response to SARS-CoV-2 variants of concern in vaccinated individuals

Matthias Becker [1,14], Alex Dulovic [1,14], Daniel Junker[1], Natalia Ruetalo[2], Philipp D. Kaiser[1], Yudi T. Pinilla[3], Constanze Heinzel[3], Julia Haering[1], Bjoern Traenkle[1], Teresa R. Wagner [1,4], Mirjam Layer[2], Martin Mehrlaender[5], Valbona Mirakaj[5], Jana Held[3,6], Hannes Planatscher[7], Katja Schenke-Layland [1,8,9,10], Gérard Krause [11,12], Monika Strengert[11,12], Tamam Bakchoul [13], Karina Althaus[13], Rolf Fendel [3,6], Andrea Kreidenweiss [3,6], Michael Koeppen[5], Ulrich Rothbauer [1,4✉], Michael Schindler [2✉] & Nicole Schneiderhan-Marra [1✉]

SARS-CoV-2 is evolving with mutations in the receptor binding domain (RBD) being of particular concern. It is important to know how much cross-protection is offered between strains following vaccination or infection. Here, we obtain serum and saliva samples from groups of vaccinated (Pfizer BNT-162b2), infected and uninfected individuals and characterize the antibody response to RBD mutant strains. Vaccinated individuals have a robust humoral response after the second dose and have high IgG antibody titers in the saliva. Antibody responses however show considerable differences in binding to RBD mutants of emerging variants of concern and substantial reduction in RBD binding and neutralization is observed against a patient-isolated South African variant. Taken together our data reinforce the importance of the second dose of Pfizer BNT-162b2 to acquire high levels of neutralizing antibodies and high antibody titers in saliva suggest that vaccinated individuals may have reduced transmission potential. Substantially reduced neutralization for the South African variant further highlights the importance of surveillance strategies to detect new variants and targeting these in future vaccines.

[1] NMI Natural and Medical Sciences Institute at the University of Tübingen, Reutlingen, Germany. [2] Institute for Medical Virology and Epidemiology, University Hospital Tübingen, Tübingen, Germany. [3] Institute of Tropical Medicine, University of Tübingen, Tübingen, Germany. [4] Pharmaceutical Biotechnology, University of Tübingen, Tübingen, Germany. [5] Department of Anaesthesiology and Intensive Care Medicine, University Hospital Tübingen, Tübingen, Germany. [6] German Center for Infection Research (DZIF), partner site Tübingen, Tübingen, Germany. [7] Signatope GmbH, Reutlingen, Germany. [8] Cluster of Excellence iFIT (EXC2180) "Image-Guided and Functionally Instructed Tumor Therapies", University of Tübingen, Tübingen, Germany. [9] Department of Women's Health, Research Institute for Women's Health, University of Tübingen, Tübingen, Germany. [10] Department of Medicine/ Cardiology, Cardiovascular Research Laboratories, David Geffen School of Medicine at UCLA, Los Angeles, USA. [11] Helmholtz Centre for Infection Research, Braunschweig, Germany. [12] TWINCORE GmbH, Centre for Experimental and Clinical Infection Research, a joint venture of the Hannover Medical School and the Helmholtz Centre for Infection Research, Hannover, Germany. [13] Institute for Clinical and Experimental Transfusion Medicine, University Hospital Tübingen, Tübingen, Germany. [14] These authors contributed equally: Matthias Becker, Alex Dulovic. ✉email: Ulrich.rothbauer@nmi.de; Michael. Schindler@med.uni-tuebingen.de; Nicole.schneiderhan@nmi.de

Since the initial outbreak in Wuhan, China in late 2019[1,2], SARS-CoV-2 has evolved into a global pandemic, with more than 138 million infections and nearly 3 million deaths (as per WHO, https://covid19.who.int/, accessed April 15, 2021), impacting severely on mental health[3,4] and global economics[5]. In response, the scientific community has made unprecedented progress, resulting in the generation of multiple vaccines, using a variety of different approaches[6–8], such as the Pfizer BNT-162b2 vaccine, which encodes a full-length trimerized spike protein[9]. In parallel, SARS-CoV-2 is continually evolving impacting its infectivity[10], transmission[11–13], and viral immune evasion[14,15]. To date, advanced genomic approaches have identified thousands of variants of SARS-CoV-2 with multiple RBD mutations circulating due to natural selection[16,17]. The variability of RBD epitopes is of specific concern as such mutations might reduce vaccine efficacy, increase viral transmission, or impair acquired immunity by neutralizing antibodies[10,18,19]. For the pandemic to be brought under control, herd immunity must be achieved through vaccination. However, there is a discourse about how long antibodies generated during the first wave persist, with some studies suggesting seroreversion between 2 and 3 months[20], while others find antibodies present for up to 7 or 8 months post infection[21–23]. Alarmingly, antibodies generated during the first wave also appear to have reduced immunoreactivity and neutralization potency toward emerging variants[22].

As the virus is known to continually mutate, particularly the emerging UK (B.1.1.7)[12], South African (B.1.351)[24], Brazil (P1)[25], Mink (Cluster 5)[26], and Southern California (hereon referred to as "LA" (B1.429)[27] variants are of concern. The UK variant has an increased risk of transmission[13] and mortality[13,28]. It further exhibits reduced neutralization susceptibility[29], which is most substantially related to a subset of RBD-specific monoclonal antibodies[14,29]. The N501Y mutation appears to mediate increased ACE2–RBD interaction[30] and is known to be critical for SARS-CoV-2 infection in vivo in mice[31]. Similarly, the South African variant, which is now spreading globally, has two escape mutations within the RBD (K417N and E484K)[24] in addition to the N501Y mutation. The combination of these three point mutations results in both a higher infection rate and reduced capacity of neutralizing antibodies produced against variants without RBD mutations of concern (hereon referred to as "wild-type")[32]. In light of these developments, and in spite of increasing data provided by vaccine companies, it remains unclear whether vaccines formulated against the original Wuhan strain of the virus will remain effective against new and emerging variants such as UK or South Africa. To understand this, we characterized the antibody response post vaccination with the Pfizer BNT-162b2 vaccine in both serum and saliva and then investigated the presence and efficacy of neutralizing antibodies against emerging variants of concern (UK, South Africa, Mink, and LA).

## Results

To analyze the humoral response generated by vaccination, SARS-CoV-2 reactive antibody titers in serum samples from vaccinated, convalescent (hereon referred to as "infected"), and uninfected (hereon referred to as "negative") individuals were measured using MULTICOV-AB[33] (Fig. 1). Descriptions of all groups of donors can be found in Supplementary Table 1. Vaccinated individuals had not been previously infected with SARS-CoV-2 as demonstrated by the absence of anti-nucleocapsid IgG and IgA (Fig. 1a). As expected, there was a typical variation in antibody titers reflecting individual immune responses (Fig. 1a, b). When comparing between vaccine doses (Fig. 1c, d), all vaccinated subjects showed an enhanced antibody response with increasing time after the first dose and a further significant boost after the second dose.

This boosting effect was so pronounced that it reached the upper limit of detection for MULTICOV-AB, as confirmed by a dilution series (Supplementary Fig. 1).

To expand our understanding of the immune response of vaccinated individuals, we analyzed their saliva for IgA and IgG antibodies. The saliva of infected and negative individuals served as controls. Infected individuals had significantly higher levels of IgA than negative ($P$ value = 0.0008) or vaccinated individuals ($P$ value = 0.03), with no significant difference seen between vaccinated and negative individuals ($P$ value = 0.23) (Fig. 2a). Conversely, the IgG response in the saliva of vaccinated individuals was significantly higher than either infected ($P$ value <0.0001) or negative individuals ($P$ value <0.0001) (Fig. 2b). These results were verified using a second antibody test measuring IgG in saliva (Supplementary Fig. 2). We also identified that vaccination with Pfizer BNT-162b2 does not appear to offer any cross-protection against other endemic coronaviruses (Supplementary Fig. 3), as seen by the absence of change in antibody titers following vaccination.

Having determined the humoral response induced by Pfizer BNT-162b2, we then examined how emerging RBD mutations present in different variants of concern impact antibody binding. For this, we included RBD mutants for the UK (501Y), South African (417N, 484K, and 510Y), Mink (453F), and LA (452R) variants in MULTICOV-AB. For the UK variant, nearly identical antibody response was observed for vaccinated and infected individuals compared with wild-type variant (Kendall's tau 0.965) (Fig. 3a). In contrast, a varied and reduced immune response was visible for the South African variant in both groups (Kendall's tau 0.844) (Fig. 3b). Both the Mink and LA variants had a similar response as the wild-type variant (Supplementary Fig. 4). Having seen that antibody-binding responses were reduced in the context of RBD mutants in the South African variant, we examined its neutralizing potential on samples from vaccinated individuals using a virus neutralization test (VNT)[34], employing a patient-derived South African variant of SARS-CoV-2. Despite the detectable variation, the VNT revealed substantially reduced neutralization for the South African variant for sera obtained from vaccinated and infected individuals (Fig. 4a). We further confirmed these findings using an ACE2 inhibition assay (Fig. 4b). Here, we additionally observed increased neutralization capacities for both wild-type (Fig. 4c) and the South African variant (Fig. 4d), in all samples derived from vaccinated individuals following the second dose, which was further confirmed by the NeutrobodyPlex[35] (Supplementary Fig. 5). Overall, our results showed individual differences in neutralization capacity for RBD mutations found in variants (Supplementary Fig. 6), with minimal to no change in neutralization for the UK, Mink, or LA variants. Conversely, neutralization capacity versus the South African variant was severely compromised.

## Discussion

RBD mutants are particularly important to track and study due to the role of the RBD:ACE2 interaction site in virus transmission and neutralization[13,30] and their potentially increased infectivity[12] or lethality[28]. This tracking has led to the identification of several variants of concern, notably the UK[12], South African[24], Mink[26], and LA[27] variants. We initiated this study to reveal the vaccine-induced immune response and most importantly to shed light on the still controversial question of how efficiently antibodies can bind and neutralize SARS-CoV-2 variants of concern. The presence of large titers of IgG antibodies within the saliva of vaccinated individuals far exceeded those seen in convalescent individuals. This was both surprising and welcome as it could indicate that vaccination might confer a

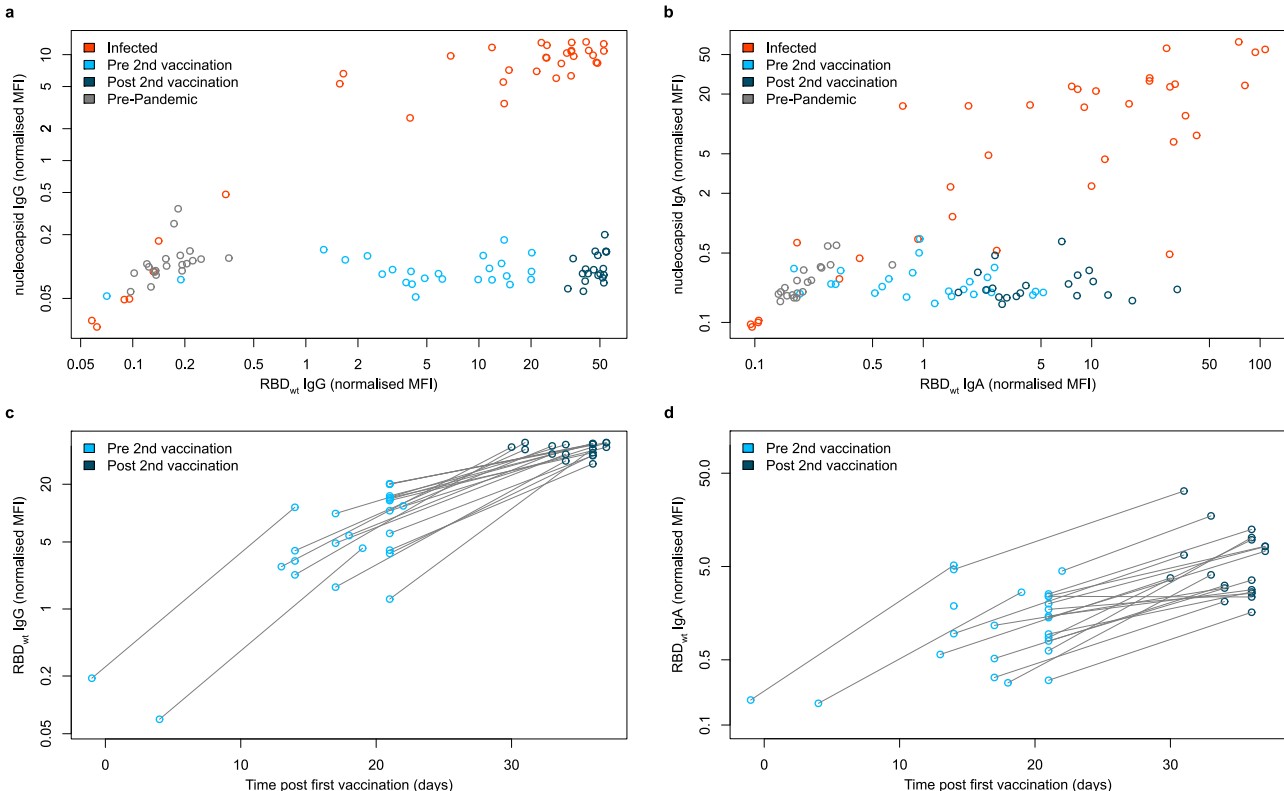

**Fig. 1 IgG and IgA response in serum samples of vaccinated, infected, and negative individuals.** IgG (**a**, **c**) and IgA (**b**, **d**) response in sera from vaccinated (pre second vaccination (light blue, $n = 25$), post second vaccination (dark blue, $n = 20$)), infected (red) ($n = 35$), and negative (gray) ($n = 20$) individuals were measured with MULTICOV-AB. IgG (**a**) and IgA (**b**) response is shown as normalized RBD wild-type (wt) versus normalized nucleocapsid MFI values allowing for visualization of separation between the different groups. Increasing antibody titers in vaccinated individuals for IgG (**c**) and IgA (**d**) is shown with increasing days post vaccination, with samples colored based on whether they are before (light blue) or after (dark blue) the second dose. Lines indicate paired samples from the same donor. Source data are provided as a Source Data file.

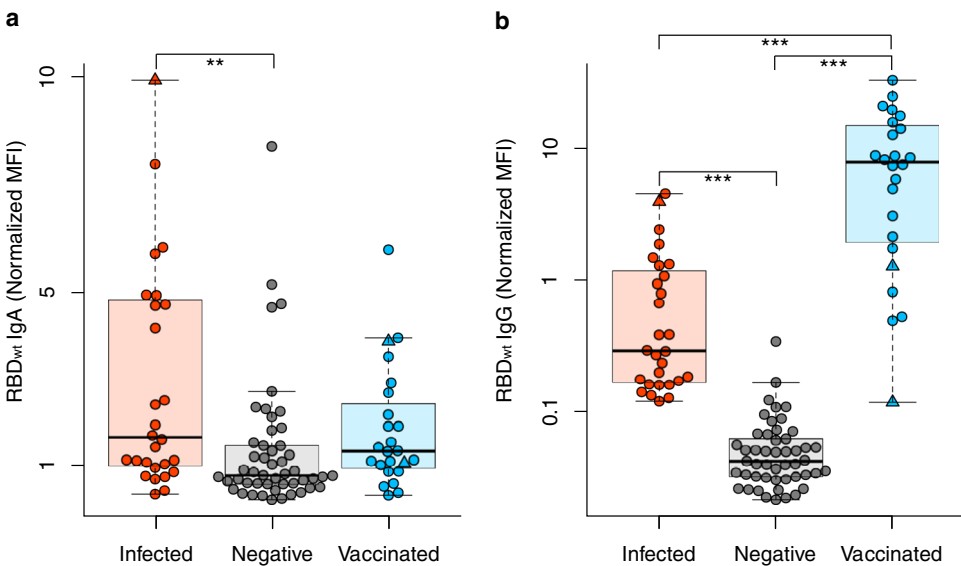

**Fig. 2 IgA and IgG response in saliva samples of vaccinated, infected, and negative individuals.** Box and whisker plots for the IgA (**a**) and IgG (**b**) response in the saliva of vaccinated (blue, $n = 22$), infected (red, $n = 26$), and negative (gray, $n = 45$) individuals. All samples were measured three times using MULTICOV-AB, normalized against QC values to remove confounding effects, and the mean calculated and displayed. Panel **b** is presented using a logarithmic scale for clarity. As additional controls, one infected and then vaccinated sample and two vaccinated samples from individuals not in contact with active COVID-19 infections are displayed as triangles. Boxes represent the median, 25th and 75th percentiles, whiskers show the largest and smallest non-outlier values. Outliers were determined by 1.5 times IQR. Statistical significance was calculated by Mann–Whitney $U$ (two-sided) with significance determined as being < 0.01. **< 0.001; ***< 0.0001. Source data are provided as a Source Data file.

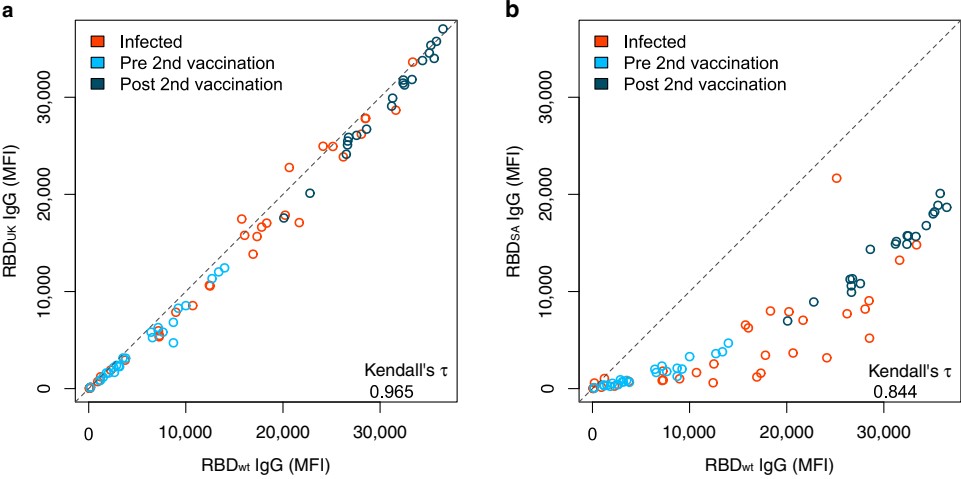

**Fig. 3 South African RBD mutant has a reduced response compared to UK RBD mutant.** UK (**a**) and South African (SA) RBD mutants (**b**) have differing effects upon antibody binding. RBD mutant antigens were generated and added to MULTICOV-AB to measure the immune response towards them in sera from vaccinated pre-second dose (light blue, $n = 25$), post second dose (dark blue, $n = 20$), and infected (red, $n = 35$) individuals, compared to the wild-type (wt) RBD. A linear curve ($y = x$) is shown as a dashed gray line to indicate an identical response between wild-type and mutant. Kendall's tau was calculated to measure the ordinal association between the mutant and wild-type. Source data are provided as a Source Data file.

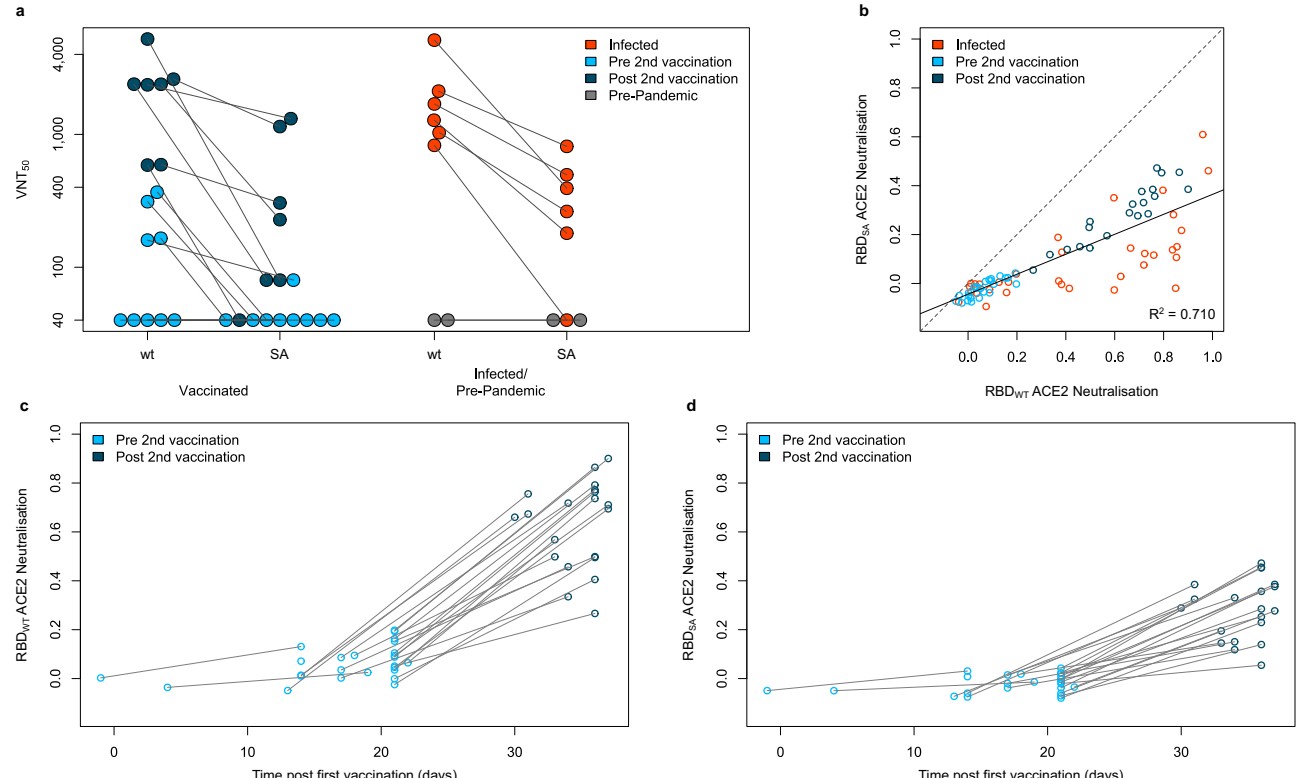

**Fig. 4 South African RBD variant has decreased neutralization compared to wild-type in vaccinated and infected samples.** Neutralization for the South African variant (SA) displayed as virus neutralizing titers ($VNT_{50}$) was measured in a virus neutralization assay compared to a wild-type variant (wt) (**a**) with sera from vaccinated (pre second vaccination (light blue, $n = 9$), post second vaccination (dark blue, $n = 7$)), infected (red, $n = 6$), and negative (pre-pandemic) (gray, $n = 2$) individuals. To confirm the reduction in neutralization seen, an ACE2 competition assay was developed and used to measure neutralization capacity for wild-type RBD (wt) and the South African RBD mutant (SA) (**b**) on sera from vaccinated (pre second vaccination (light blue, $n = 25$), post second vaccination (dark blue, $n = 20$)), infected (red, $n = 35$) and negative (pre-pandemic, gray, $n = 20$) individuals. 0 indicates that no neutralization is present while 1 indicates maximum neutralization. A linear regression ($y = -0.044 + 0.408x$) for all samples is shown in gray with the $R^2$ included. When examining vaccinated samples only, wild-type neutralization (**c**) is significantly increased following the second vaccine dose. For South African neutralization (**d**), while it is increased following the second dose, there is a significant reduction when compared to wild-type. Lines in (**a**), (**c**), and (**d**) indicate paired samples from the same donor. Source data are provided as a Source Data file.

sterilizing immune response in the oral cavity and thereby lower virus transmission. Focusing on antibody response, we examined in detail the effects of RBD mutations observed in emerging variants of concern. While only minor differences were detectable for the UK, Mink, and LA variants, a substantial reduction in RBD-binding antibodies was observed for the South African variant. These findings were confirmed at the functional level by a VNT using patient-derived viral isolates, showing a significant decrease in the neutralizing capacity of sera from vaccinated or infected individuals for the South African variant, in accordance with other recently published studies[36–39]. This could provide a reasonable explanation for infection in vaccinated or convalescent individuals with the South African variant and suggests a potential reduction in the efficacy of the Pfizer BNT-162b2 vaccine from a B-cell perspective.

This study is limited by the sample size, the restriction to only two time points after vaccination for data analysis, and the lack of paired saliva and serum samples for our infected and negative groups. However, we want to note that our sample size examined is similar to or larger than most other sample sets used to examine immune response to mutants in detail[40] and that we examined paired serum and saliva samples for all of our vaccinated subjects. Furthermore, we performed VNT assays comparing wild-type to the South African variant with authentic virus isolates on human cells, in contrast to utilizing a pseudotype neutralization assay[40] or genetically engineered wild-type variant[37]. Notably, we focused our study on a detailed characterization of the humoral immune response, as a proxy of an individual's immune response, as T-cell immunity has already been extensively studied[21,41]. Future work should investigate the antibody response and their persistence from different vaccines (i.e., AstraZeneca-Oxford, Moderna) against similar or newly emerging RBD mutants over a longer timeframe with increased sample size.

Viewed in a larger context, the impaired RBD-binding capacity to mutations in emerging variants of concern highlights the importance of updating current vaccines accordingly. Particular attention should be paid to the South African variant, because of the reduced neutralizing potency identified, while spontaneous independent enrichments of the E484K mutation have been observed in several countries[42].

## Methods

**Data reporting**. No statistical analysis was used to determine the sample size. The experiments were not randomized, and the investigators were not blinded during either experimentation or data analysis.

**Study recruitment, sample collection, and ethics statement**. Serum and saliva samples were collected from healthcare workers, vaccinated with the Pfizer BNT-162b2 vaccine. Two serum samples were collected from each individual. The first collection took place on average 17 days following administration of the first dose, while the second collection took place on average 12 days following administration of the second dose. Saliva samples were collected 7–10 days following the second serum sampling. Infected serum and saliva samples were collected from different groups of individuals. Serum samples were collected from individuals hospitalized at Universität Klinikum Tübingen between March 25, 2020 and January 22, 2021. All individuals tested positive for SARS-CoV-2 by PCR. Saliva samples were collected from individuals who had previously been infected with SARS-CoV-2 between March 7, 2020 and June 19, 2020. All individuals had previously tested positive by PCR, or were confirmed as being previously infected by ELISA measurements plus the presence of at least one key symptom (i.e., coughing, fever). Negative serum and saliva samples were also from different groups of individuals. Negative serum samples were purchased from Central Biohub. All of these samples were collected pre-pandemic. Negative saliva samples were collected at the Institute of Tropical Medicine, Universität Tübingen from negative individuals and were confirmed to be anti-SARS-CoV-2 negative by serum ELISA. As additional controls, serum and saliva samples were collected from two individuals vaccinated with Pfizer BNT-162b2, who do not have contact with active SARS-CoV-2 infected patients, and one individual who had been previously infected with SARS-CoV-2 and was later vaccinated.

For serum collection, blood was taken by venipuncture, the serum extracted, and then frozen at −80 °C until use. For saliva collection, all individuals spat directly into a collecting tube. To inactive the samples, TnBP and Triton X-100 were added to final concentrations of 0.3% and 1×, respectively. Saliva samples were then frozen at −80 °C until further use.

This study was approved by the Ethics Committee of Eberhard Karls University Tübingen and the University Hospital Tübingen under the approval number 222/2020BO2 to Dr. Karina Althaus, Institute for Clinical and Experimental Transfusion Medicine, University Hospital Tübingen, and 312/2020BO1 (Coro-Buddy) to Dr. Andrea Kreidenweiss, Institute of Tropical Medicine, University Hospital Tübingen and Eberhard Karls University Tübingen. All participants gave written informed consent. Characteristics of vaccinated and infected serum donors can be found in Supplementary Table 1. Characteristics of vaccinated, infected, and negative saliva donors can be found in Supplementary Table 2.

**Expression of RBD mutants**. The pCAGGS plasmid encoding the receptor-binding domain (RBD) of SARS-CoV-2 was kindly provided by F. Krammer[43]. RBDs of SARS-CoV-2 variants of concern were generated by PCR amplification of fragments from wild-type DNA template followed by fusion PCRs to introduce described mutation N501Y for the UK variant and additional mutations K417N and E484K for the South African variant[28,44]. Forward primer RBDfor and reverse primer N501Yrev were used for amplification of fragment 1, forward primer N501Yfor and reverse primer RBDrev were used for amplification of fragment 2. Both fragments containing an overlap sequence at the 3′ and 5′ end were fused by an additional PCR using forward primer RBDfor and RBDrev. Based on cDNA for the UK variant, additional mutations of the South African Variant were introduced by PCR amplification of three fragments using forward primer RBD-for and reverse primer K417Nrev, forward primer K417Nfor and reverse primer E484Krev, forward primer E484Kfor and reverse primer RBDrev. Amplified fragments were assembled by subsequent fusion PCR using forward primer RBDfor and RBDrev. RBD mutation L452R as recently reported for the SARS-CoV-2 variant of concern identified in Southern California (referred to in this manuscript as "LA"), was introduced using primer RBD-for and reverse primer L452Rrev for amplification of fragment 1 and forward primer L452Rfor for fragment 2. Both fragments were subsequently fused using primers RBDfor and RBDrev. DNA coding for mutant RBDs (amino acids 319–541 of respective spike proteins) was cloned into Esp3I and EcoRI site of pCDNA3.4 expression vector with the N-terminal signal peptide (MGWTLVFLFLLSVTAGVHS) for the secretory pathway that comprises Esp3I site. All expression constructs were verified by sequence analysis. A full list of primers used in this study can be found in Supplementary Table 3. Confirmed constructs were expressed in Expi293 cells[33]. Briefly, cells were cultivated (37 °C, 125 rpm, 8% (v/v) $CO_2$) to a density of $5.5 \times 10^6$ cells/mL, diluted with Expi293F expression medium and transfection of the corresponding plasmids (1 µg/mL) with expifectamine as per the manufacturer's instructions. 20 h post transfection enhancers were added as per the manufacturer's instructions. Cell suspensions were then cultivated for 2–5 days (37 °C, 125 rpm, 8 % (v/v) $CO_2$) and then centrifuged (4 °C, 23,900×g, 20 min) to clarify the supernatant. Supernatants were then filtered with a 0.22-µm membrane (Millipore, Darmstadt, Germany) and supplemented with His-A buffer stock solution (20 mM Na2HPO4, 300 mM NaCl, 20 mM imidazole, pH 7.4). The solution was then applied to a HisTrap FF crude column on an Äkta pure system (GE Healthcare, Freiburg, Germany), extensively washed with His-A buffer, and eluted with an imidazole gradient (50–400 mM). Eluted proteins were dialyzed against PBS.

**Bead coupling**. Coupling of RBD mutant antigens was done by Anteo coupling (#A-LMPAKMM-10, Anteo Tech Reagents) following the manufacturer's instructions. Briefly, 100 µL of spectrally distinct populations of MagPlex beads ($1.25 \times 10^6$) (Luminex) were activated in 100 µL of AnteoBind Activation Reagent for 1 h at room temperature. The activated beads were washed twice with 100 µL of Coupling Buffer using a magnetic separator. Following this, 50 µg/mL of antigen (diluted in Coupling buffer) was added to the beads and incubated for 1 h at room temperature. The beads were then washed twice with 100 µL Coupling buffer and blocked for 1 h at room temperature in 0.1% BSA in Coupling Buffer. After washing twice in storage buffer, the beads were stored at 4 °C until further use.

**MULTICOV-AB**. MULTICOV-AB[33], a multiplex immunoassay that simultaneously analyses 20 antigens was performed on all samples. In addition to the antigens presently included in MULTICOV-AB (Supplementary Table 4), RBD mutants from variants of concern were also included for measurements of all serum samples. All RBD mutants except the Mink variant (#40592-V08H80, Sino Biological) were produced in-house. Both IgG and IgA were measured for all serum samples. To adapt MULTICOV-AB to analyze antibodies in saliva, the dilution factor was changed from 1:400 for the serum to 1:12 for saliva. Samples were diluted into assay buffer (1:4 Low Cross Buffer (Candor Bioscience GmBH) in CBS (1× PBS + 1% BSA) + 0.05% Tween20) inside a sterile workbench. In total, 25 µL of diluted sample was then added to 25 µL of 1× Bead Mix[33] using a 96-well plate (#3600, Corning). Samples were incubated for 2 h at 20 °C, 750 rpm on a Thermomixer (Eppendorf), after which the unbound antibodies were removed by washing three times with Wash Buffer (1× PBS, 0.05% Tween20) using a

microplate washer (Biotek 405TS, Biotek Instruments GmBH). Bound antibodies were detected using either 3 μg/mL RPE-huIgG (#109-116-098, Dianova) or 5 μg/mL RPE-huIgA (#109-115-011, Dianova) by incubation for 45 min at 20 °C, 750 rpm on a thermomixer. Following another washing step, beads were re-suspended in 100 μL of Wash Buffer and re-shaken for 3 mins at 20 °C, 1000 rpm. Plates were then measured using a FLEXMAP3D instrument (Luminex) using the following settings: 80 μL (no timeout), 50 events, Gate:7500–15000 and Report Gain:Standard PMT. Each sample was measured in three independent experiments. Three cut-off (CO) samples with a known MFI value were generated as in ref. [45] and included on each plate as a quality control. CO2 was used to generate the plate-by-plate CO value for the IgG Spike and wtRBD of SARS-CoV-2, with CO3 being used for the same purpose but for IgA. Raw median fluorescence intensity (MFI) values were divided by the mean MFI of CO2 (for IgG) or CO3 (for IgA), included on each plate to produce a normalization value. For serum measurements of the Spike and wtRBD of SARS-CoV-2, a normalized MFI value > 1 indicates positivity.

**Saliva IgG ELISA.** To validate saliva measurements by MULTICOV-AB, samples were re-measured using an in-house ELISA established by the Institute of Tropical Medicine, Universität Tübingen. Saliva samples were analyzed for SARS-CoV-2 wild-type RBD reactive IgG antibodies by an in-house ELISA developed at the Institute of Tropical Medicine, University of Tübingen. SARS-CoV-2 RBD recombinant protein was dissolved in PBS to a final concentration of 2 μg/mL. In all, 50 μL was then coated into 96-well Costar microtiter high binding plates (#3590, Corning) and blocked at 4 °C overnight with The Blocking Solution (Candor Bioscience GmbH), at room temperature on a microplate shaker set to 700 rpm. Before each of the following steps, wells were washed with PBS/0.1% Tween20. Saliva samples were diluted using The Blocking Solution (1:3–1:729) and 100 μL added to each well. Plates were then incubated at room temperature for 1 h. For detection, 1:20,000 biotinylated anti-human IgG (#109-065-008, Jackson Immuno Research Laboratories) and 1:20,000 Streptavidin-HRP (#109-035-098) were added and incubated for 1 h and 30 min, respectively. For visualization, TMB was added and the reaction was stopped using 1 M HCl. The plate was read at 450 nm and 620 nm using a microplate reader (CLARIOstar, BMG LABTECH, running Software Version 5.40 R2). To estimate the concentration of the IgG antibodies in the saliva, a dilution series of highly pure human IgG (#31154, ThermoFisher) was in parallel coated on the ELISA plates and quantified using the secondary antibodies. A four-parameter logistic curve using MARS Data Analysis Software Version 3.31 was fitted to the respective OD values and the antibody concentrations in the sample specimen calculated[46].

**Neutralization assays.** Viral neutralization assays[34] for the wild-type (Tü1) variant and the South African variant were performed on 16 vaccinated, 6 infected, and 2 negative serum samples. Briefly, Caco-2 cells were cultured at 37 °C with 5% $CO_2$ in Dulbecco's modified Eagle medium (DMEM), supplemented with 10% fetal calf serum (FCS), 2 mM L-glutamine, 100 mg/mL penicillin–streptomycin, and 1% non-essential amino acids (NEAA). The clinical isolate (200325_Tü1)[34] which belongs to the lineage B.1.126 is referred to as "wild-type" in this manuscript. The South African variant (210211_SaV) was isolated from a throat swab collected in January 2021 at the Institute for Medical Virology and Epidemiology of Viral Diseases, University Hospital Tübingen, from a PCR-positive patient. In total, 100 μL of patient material was diluted in medium and used to directly inoculate 150,000 Caco-2 cells in a six-well plate. At 48 h post infection, the supernatant was collected, centrifuged, and stored at −80 °C. After two consecutive passages, the supernatant was tested by qRT-PCR confirming the presence of three point mutations (N501Y, K417N, and E484K). NGS confirmed that the clinical isolate belongs to the lineage B.1.351. Sequence comparison of the Spike protein of 200325_Tü1 and 210211_SaV can be found as Supplementary Data 1. Caco-2 cell infection with 210211_SAv was detected by western blotting, using sera from a convalescent patient. The multiplicity of infection determination (MOI) was conducted by titration using serial dilutions of both virus stocks. The number of infectious virus particles per millimeter was calculated as (MOI x cell number)/(infection volume), where MOI = -ln (1 – infection rate). For neutralization experiments, $1 \times 10^4$ Caco-2 cells/well were seeded in 96-well plates the day before infection in a medium containing 5% FCS. Cells were co-incubated with SARS-CoV-2 clinical isolate 200325_Tü1 or SARS-CoV-2 clinical isolate 210211_SAv at an MOI of 0.7. Patient sera were added in serial twofold dilutions from 1:40 to 1:5120. At 48 h post infection, cells were fixed with 80% acetone for 5 min, washed with PBS, and blocked for 1 h at room temperature (RT) with 10% normal goat serum (NGS). Cells were incubated for 1 h at RT with 100 μL of serum from a hospitalized convalescent donor in a 1:10,000 dilution and washed three times with PBS. In total, 100 μL of goat anti-human Alexa594 (1:2000) in PBS was used as a secondary antibody for 1 h at RT. Cells were then washed three times with PBS and counterstained with 1:20,000 DAPI solution (2 mg/mL) for 10 min at RT. For quantification of infection rates, images were taken with the Cytation3 (BioTek) and DAPI-positive and Alexa594-positive cells were automatically counted by the Gen5 software (BioTek). Virus neutralizing titers (VNT$_{50S}$) were calculated as the half-maximal inhibitory dose (ID$_{50}$) using 4-parameter nonlinear regression (GraphPad Prism). An overview of the VNT assay and examples of cells treated with both variants from one vaccinated and one infected individual's serum can be found in Supplementary Fig. 7.

**ACE2 competition assay.** Biotinylated recombinant human ACE2 (#10108-H08H-B, Sino Biological) was diluted to a final concentration of 571.4 ng/mL in assay buffer (1:4 Low Cross Buffer (Candor Bioscience GmbH) in CBS (1× PBS + 1% BSA) + 0.05% Tween20) to create ACE2 buffer. Plasma samples were diluted 1:50 in assay buffer and then to a final concentration of 1:400 in ACE2 buffer under a sterile workbench, resulting in a final ACE2 concentration of 500 ng/mL in all samples. In all, 25 μL of diluted plasma samples were then added to 25 μL of 1× BeadMix[33] per well of a 96-well plate (#3642, Corning). In addition to the standard bead mix used in MULTICOV-AB, all bead coupled RBD mutants were included. As a control, 500 ng/mL ACE2 was also used. Samples were incubated at 21 °C, 750 rpm for 2 h on a thermomixer. Samples were then washed using a microplate washer with Wash Buffer (1× PBS + 0.05% Tween20). 30 μL of 2 μg/mL Streptavidin-PE (#SAPE-001, Moss) was added to each well and incubated for 45 min, 21 °C, 750 rpm in darkness on a thermomixer. Following incubation, samples were washed again and then resuspended in 100 μL of Wash Buffer, before being shaken again for 3 min at 1000 rpm. Samples were read individually on a FLEXMAP3D instrument under the following settings: 80 μL (no timeout), 50 events, Gate: 7500–15,000, Reporter Gain: Standard PMT. MFI values were then normalized against the control wells. All samples were measured in duplicates and the mean is reported.

**Neutrobody Plex.** For further validation of the neutralization response, all samples were analyzed using the NeutrobodyPlex[35] using a final Nanobody (NMI1267) concentration of 2 nM. Samples were diluted 1:25 into assay buffer (same as MULTICOV-AB) and then a further 1:8 in neutrobody buffer (MULTICOV-AB assay buffer + 4.56 nM nanobody). In total, 25 μL of diluted sample was then added to 25 μL of 1× MULTICOV-AB Bead Mix[33] using a 96-well plate (#3600, Corning). Samples were incubated for 2 h at 21 °C, 750 rpm on a thermomixer (Eppendorf), after which the unbound antibodies were removed by washing three times with Wash Buffer (1× PBS, 0.05% Tween20) using a microplate washer (Biotek 405TS, Biotek Instruments GmbH). Bound antibodies were detected using 3 μg/mL RPE-huIgG (#109-116-098, Dianova) by incubation for 45 mins at 21 °C, 750 rpm on a thermomixer. Following another washing step, beads were re-suspended in 100 μL of Wash Buffer and re-shaken for 3 min at 21 °C, 1000 rpm. Plates were then measured using a FLEXMAP3D instrument (Luminex) using the same settings as in MULTICOV-AB above. Samples were measured on the same plate with their respective MULTICOV-AB IgG measurement. In addition to MULTICOV-AB controls, for plate-to-plate qualification one control sample was processed on all plates.

**Data analysis.** Data analysis and figure generation were performed in RStudio (Version 1.2.5001), running R (version 3.6.1) with the additional packages "bees-warm" and "RcolorBrewer" for data depiction purposes only. The type of statistical analysis performed (when appropriate) is listed in the figure legends. Figures were generated in Rstudio and then edited for clarity in Inkscape (Inkscape 0.92.4). Mann–Whitney $U$ test was used to determine the difference between signal distributions from different sample groups using the "wilcox.test" function from R's "stats" library. Kendall's $\tau$ coefficient was calculated in order to determine the ordinal association between the observed antibody responses towards RBD mutant and wild-type proteins using the "cor" function from R's "stats" library. Linear regression was performed to assess the reduction in ACE2 neutralization observed for RBD mutants compared to wild-type proteins using the "lm" function from R's "stats" library. Pre-processing of data such as matching sample metadata and collecting results from multiple assay runs was performed in Excel 2016. GraphPad Prism version 8.4.0 was used to process VNT data.

**Reporting summary.** Further information on research design is available in the Nature Research Reporting Summary linked to this article.

## Data availability

Source data are provided with this paper.

## Code availability

Custom analysis code in R and required input files have been deposited on GitHub: https://github.com/BeckerMatthias/Vaccination_VoC_Publication.

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

## Acknowledgements

We thank Johanna Griesbaum for technical assistance, Florian Krammer for providing us with expression plasmids for the Spike Trimer and RBD, Tina Ganzenmüller and the diagnostic department of the Institute for Medical Virology, University Hospital Tübingen for providing patient samples, and the NGS competence center of the University Hospital Tübingen for rapid sequencing. This work was financially supported by the State Ministry of Baden-Württemberg for Economic Affairs, Labour and Housing Construction (grant numbers FKZ 3–4332.62-NMI-67 and FKZ 3–4332.62-NMI-68), the Ministerium für Wissenschaft und Kunst Baden-Württemberg, the Initiative and Networking Fund of the Helmholtz Association of German Research Centres (grant number SO-96), the Deutsche Forschungsgemeinschaft (DFG-KO 3884/5-1) and the EU Horizon 2020 research and innovation programme (grant agreement number 101003480—CORESMA). The funders had no role in study design, data collection, data analysis, or the decision to publish.

## Author contributions

U.R., M.K., M.B., A.D., and N.S.M. conceived the study. M.B., A.D., Ja.H., R.F., A.K., Mi.S., U.R., and N.S.M. designed the experiments. K.S.L., Mo.S., G.K., Mi.S., M.K., and N.S.M. procured funding. M.B., A.D., D.J., N.R., C.H., Ju.H., and M.L. performed experiments. Y.T.P., M.M., V.M., T.B., K.A., R.F., A.K., M.K., U.R., and N.S.M. collected samples or organized their collection. P.D.K., B.T., and T.W. produced the RBD mutants. H.P., M.B., and D.J. produced and analyzed the QC samples in MULTICOV-AB. M.B., A.D., N.R., and Mi.S. performed the data analysis. M.B., A.D., D.J., and N.S.M. generated the figures. A.D. wrote the first draft of the manuscript with input from M.B., D.J., Ja.H., K.S.L., Mo.S., R.F., A.K., Mi.S., U.R., and N.S.M. A.D., M.B., R.F., Mi.S., U.R., and N.S.M. revised the manuscript. All authors approved the final version of the manuscript.

## Competing interests

T.R.W., P.K., N.S.M., and U.R. are named as inventors on a patent application (EP 20 197 031.6) claiming the use of the described Nanobodies used in the NeutrobodyPlex for diagnosis and therapeutics filed by the Natural and Medical Sciences Institute. N.S.-M. was a speaker at Luminex user meetings in the past. The Natural and Medical Sciences Institute at the University of Tuebingen is involved in applied research projects as a fee for services with Luminex. The remaining authors declare no competing interests.
