## [Peer Review File · Nature Communications]

Reviewers' Comments:

Reviewer #1:

Remarks to the Author:

This is a timely and solid article on measurement of humoral responses to COVID-19 vaccination. The work builds on the multiplexed assay published by the same group in 2020. Here, the authors have extended the assay to measure antibodies against mutated RBD proteins. They have also developed a proxy assay for neutralizing antibodies based on inhibition of ACE2 binding to RBD. The results are convincing and support the conclusion that vaccinated individuals have high levels of IgG antibodies in saliva. This is an important observation that supports the view that vaccination results in sterilizing immunity. This is currently an ongoing discussion, and this work is therefore an important contribution.

One can always wish for more data. One would for example expect that the authors have a large database with results obtained from sero-prevalence screening. Yet, the results here are quite convincing to the extent that they show very high levels of antibodies following vaccination, and the new results on IgG in saliva are important enough to be published as they are.

I would ask the authors to provide information in the data tables on the different types of samples. It seems as if samples CO1 CO2 and CO3 are replicates. What are these samples?

Reviewer #2:

Remarks to the Author:

In this manuscript, sera and saliva from people vaccinated with Pfizer BNT-162b2 vaccine were tested for binding IgG and IgA antibodies post vaccination and compared to convalescent and uninfected sera. The authors find that vaccination preferentially induces IgG in saliva as opposed to the IgA induced by infection. Binding titers were measured against a series of antigens including RBD mutants. The data demonstrates a reduction in binding against SA variant but not others, and this was confirmed by a reduction in neutralization titer. Overall, the main data points have already been published as in "SARS-CoV-2 spike variants exhibit differential infectivity and neutralization resistance to convalescent or post-vaccination sera", but these other publications are not referenced. Another main critique is that the methods are not clearly written for others to reproduce. There are many homemade assays, and there are no assay controls included in the dataset.

-Data is "normalized against QC values to remove confounding effects"" but what are the QC values?

-There are positive signals in many of the 'negative' samples but no definition of positivity is given.

-In extended data Figure 1, the legend says, "A uniform curve shape for all samples (including the negative control) confirmed reliability of the generated data". Not sure how the shape of the data confirms reliability?

-In extended data Figure 1, the sera could just be further diluted to generate a true curve.

In extended data 2, the methods list that data is concentrations "as estimated by a respective dilution series of highly pure human IgG" but it's not clear how IgG can be used as a standard for a virus-specific IgG (in a plate coated with virus protein)? Or how the estimate was calculated.

-How different are the sequences in the S protein in the two isolates used for neutralization?

-Very hard to see the clear circles in Fig. 2.

Reviewer #1

Comment #1

This is a timely and solid article on measurement of humoral responses to COVID-19 vaccination. The work builds on the multiplexed assay published by the same group in 2020. Here, the authors have extended the assay to measure antibodies against mutated RBD proteins. They have also developed a proxy assay for neutralizing antibodies based on inhibition of ACE2 binding to RBD. The results are convincing and support the conclusion that vaccinated individuals have high levels of IgG antibodies in saliva. This is an important observation that supports the view that vaccination results in sterilizing immunity. This is currently an ongoing discussion, and this work is therefore an important contribution. One can always wish for more data. One would for example expect that the authors have a large database with results obtained from sero-prevalence screening. Yet, the results here are quite convincing to the extent that they show very high levels of antibodies following vaccination, and the new results on IgG in saliva are important enough to be published as they are.

Author response: We thank the reviewer for these kind comments on the manuscript

Comment #2

I would ask the authors to provide information in the data tables on the different types of samples. It seems as if samples CO1 CO2 and CO3 are replicates. What are these samples?

Author response: CO1, CO2 and CO3 samples are QC samples included on every plate for quality control purposes. CO2 is used to generate the plate-by-plate cut off for the Spike and RBD of SARS-CoV-2 for IgG, with CO3 being used for the same for IgA. The CO samples themselves consist of a mixture of serum samples with a known MFI value. Their generation for assays has been described in a previous publication by the group. A few sentences explaining their generation and use has now been added to the Methods section for MULTICOV-AB.

Reviewer #2

Comment #1

In this manuscript, sera and saliva from people vaccinated with Pfizer BNT-162b2 vaccine were tested for binding IgG and IgA antibodies post vaccination and compared to convalescent and uninfected sera. The authors find that vaccination preferentially induces IgG in saliva as opposed to the IgA induced by infection. Binding titers were measured against a series of antigens including RBD mutants. The data demonstrates a reduction in binding against SA variant but not others, and this was confirmed by a reduction in neutralization titer. Overall, the main data points have already been published as in “SARS-CoV-2 spike variants exhibit differential infectivity and neutralization resistance to convalescent or post-vaccination sera”, but these other publications are not referenced.

Author response: We thank the reviewer for this comment. At the time of writing, we unfortunately missed the explicit reference stated above as we made our initial submission to a different journal at the same time that this publication was placed on medRxiv. We have now gone back through the literature and cited similar studies that have been published since our initial submission, including short statement stating that our findings are in

accordance with others, as well as updating references to pre-print and published studies that have now been published.

Comment #2

Another main critique is that the methods are not clearly written for others to reproduce. There are many homemade assays, and there are no assay controls included in the dataset.

Author response: We have tried to clarify some aspects of the methods section, however we would like to point out that all assays other than the non-bead based Saliva IgG assay are previously published and therefore have full in depth protocols available in their original publications. We also disagree with the label of homemade assays, as all assays other than the non-bead based in-house IgG saliva assay have been previously fully validated and published, with MULTICOV-AB and NeutrobodyPlex being commercially available. We have ensured that all published and validated assays are thoroughly cited to direct the reader to the appropriate publication. All assays did include controls which was stated in the extended methods section. Control values are included in the MULTICOV-AB data file as these are required in order to set the Cut Off values in order to determine positivity. As control samples for all other assays were used only to determine whether a plate passed or failed, we did not include them in the dataset. Should the reviewer and editor request so, we will happily provide data for all control samples.

Comment #3

Data is “normalized against QC values to remove confounding effects” but what are the QC values?

Author response: Please see previous comment to reviewer 1.

Comment #4

There are positive signals in many of the ‘negative’ samples but no definition of positivity is given.

Author response: We thank the reviewer for this comment. Positivity can only be defined for serum samples due to an inability to generate Cut off values for saliva. As a result, normalised MFI was used to show all data so that it is easily comparable to the reader. For clarification, an additional sentence has been added to the methods stating “for serum measurements of the Spike and wtRBD of SARS-CoV-2, a normalized MFI value >1 indicates positivity”. A more detailed overview of how MULTICOV-AB determines positivity can be found in the previous publication (Becker et al, 2021, Nat. Comms) which is referenced throughout the manuscript.

Comment #5

In extended data Figure 1, the legend says, “A uniform curve shape for all samples (including the negative control) confirmed reliability of the generated data”. Not sure how the shape of the data confirms reliability?

Author response: The experiments shown in Extended Data Figure 1 were performed to show the presence of the plateau. The shape of the data lines being consistent confirms the presence of the plateau. By reliability, we meant that the curve was the same shape for all samples.

Comment #6

In extended data Figure 1, the sera could just be further diluted to generate a true curve.

Author comment: We thank the reviewer for this comment. While the data could be diluted further to generate a true curve, we feel that the dilution series currently used is sufficient to demonstrate the point we wish to make. We would like to point out to reviewer that a wide range of samples were chosen to confirm this linearity, resulting in us using a log curve to display the data. This has now been appropriately labelled. We would also like to point out that the current volume of serum used in this assay is 0.06 μ L for the 1:400 dilution to 0.001 μ L for the 1:12800 dilution. This is nearing the lower limit of detection for the assay as previously stated in the original MULTICOV-AB publication.

Comment #7

In extended data 2, the methods list that data is concentrations “as estimated by a respective dilution series of highly pure human IgG” but it’s not clear how IgG can be used as a standard for a virus-specific IgG (in a plate coated with virus protein)? Or how the estimate was calculated.

Author response: We thank the reviewer for this comment. We have now expanded in the methods section how this IgG was used as a standard and how it was calculated. The use IgG as a standard is based of another recently accepted publication which is now cited in the manuscript.

Comment #8

How different are the sequences in the S protein in the two isolates used for neutralization?

Author response: We thank the reviewer for this comment. Differences between the two isolates are now shown as Supplementary File 1 and a sentence has been added to the Methods section to illustrate this. The confirmation of the correct SA mutations in the SA isolate was already listed within the manuscript.

Comment #9

Very hard to see the clear circles in Fig. 2.

Author response: We thank the reviewer for this comment. We have now displayed the clear circles as triangles and have updated the figure legend accordingly. We have also made the same alteration to Extended Data Fig 2 as this used the same clear circles within a boxplot. We hope that this improves the legibility of the figure.